# Noisy Self-Training with Synthetic Queries for Dense Retrieval

**Fan Jiang** and **Tom Drummond** and **Trevor Cohn** [*]
School of Computing and Information Systems
The University of Melbourne, Victoria, Australia
fan.jiang1@student.unimelb.edu.au
{tom.drummond, trevor.cohn}@unimelb.edu.au

## Abstract

Although existing neural retrieval models reveal promising results when training data is abundant and the performance keeps improving as training data increases, collecting high-quality annotated data is prohibitively costly. To this end, we introduce a novel noisy self-training framework combined with synthetic queries, showing that neural retrievers can be improved in a self-evolution manner with no reliance on any external models. Experimental results show that our method improves consistently over existing methods on both general-domain (e.g., MS-MARCO) and out-of-domain (i.e., BEIR) retrieval benchmarks. Extra analysis on low-resource settings reveals that our method is data efficient and outperforms competitive baselines, with as little as 30% of labelled training data. Further extending the framework for reranker training demonstrates that the proposed method is general and yields additional gains on tasks of diverse domains.[1]

## 1 Introduction

As an important task, information retrieval (IR) refers to the task of finding relevant texts from a large collection of passages or documents to satisfy the specific information needs of users. The information need is usually expressed as a short textual query, and the task is formulated as retrieving texts that are most relevant to the given query.

Recently, impressive achievements have been made in neural retrieval models through adopting large-scale pre-trained models (Devlin et al., 2019). Dual-encoders typically serve as the backbone architecture, which enables retrieving relevant knowledge from collections with millions or billions of passages in a fraction of time (Karpukhin et al., 2020). Unlike traditional term-matching-based lexical retrievers, such as TF-IDF (Manning et al.,

---

[*]Now at Google DeepMind

[1]Source code is available at https://github.com/Fantabulous-J/Self-Training-DPR

2008) or BM25 (Robertson and Zaragoza, 2009), which can be applied without any training, neural retrieval methods normally require training on a sufficient number of human-labelled query-passage pairs to work well. Nevertheless, due to the high cost of human annotations, the number of available query-passage pairs is way smaller compared to the size of passage collections (500k at most (Bajaj et al., 2016) v.s. 21M passages (Kwiatkowski et al., 2019)). Moreover, the situation becomes increasingly worse in out-of-domain applications, where only a few or no training examples are available. Directly applying neural retrieval models trained on high-resource datasets typically achieves low out-of-domain performance and lags behind their lexical counterparts (Thakur et al., 2021).

In this work, we aim to improve the performance of state-of-the-art neural retrieval models on both general domain and out-of-domain datasets, by using automatically synthesised queries. For this purpose, we use a query generator to automatically synthesise queries for each passage in the target dataset, which we then use for pre-training, based on a self-training objective (Scudder, 1965). More specifically, instead of directly training the model on synthetic query-passage pairs (Ma et al., 2021), we use a neural retrieval model that was trained on the labelled dataset as the teacher to generate soft labels for each synthetic query, providing a supervision signal which is more robust to noise in the generated data. This ameliorates issues with the synthetic queries which are often generic or ambiguous, meaning that the query is only weakly related to its originating passage, and accordingly, other passages may match the synthetic query equally well or better. Table 1 shows a motivating example, where the synthetic query is spuriously-related to its originating passage with only the keyword *achievement* matched. However, the top-2 sampled "negatives" are highly related to the query in topics and semantics, according to both human judgment

| Synthetic Query: what was the major achievement of the manhattan project? | | |
|---|---|---|
| **Originating Passage** | **Logits** | **Hard** |
| ...The only cloud hanging over the impressive **achievement** of the atomic researchers and engineers is what their success truly meant; hundreds of thousands of innocent lives obliterated. | 0.14 | 1 |
| **Sampled Negatives** | | |
| ...success of the 20th century was the Manhattan Project. The Manhattan Project **assimilated concepts and leaders from all scientific fields and engineering disciplines to construct the first two atomic bombs**. | 1.0 | 0 |
| The Manhattan Project was an epic, secret, wartime effort to design and build the world's first nuclear weapon. ... the $20 billion project **resulted in the production of the first uranium and plutonium bombs.** | 0.92 | 0 |
| The Manhattan Project is an American film, released in 1986. The plot revolves around a gifted high school student who decides to construct an atomic bomb for a national science fair. | 0.4 | 0 |

Table 1: A motivating example for the issue of weak-relation between synthetic query and originating passages , where top-2 sampled 'negatives' are way more relevant to the query than its originating passage. Negatives and associated Logits are from a dense model fine-tuned on MS-MARCO and are normalised for clarity.

and model predictions. By contrast, simply taking the originating passage as positive and equally treating all negatives with hard labels regardless of their relevance to the query will completely mislead the model, leading to poor semantic matching patterns between query and passages.

Furthermore, to prevent the student from blindly imitating the teacher's behaviour, we pollute its inputs either in the query or passage by injecting noise. This ensures that the student has a different view on inputs compared to the teacher, encouraging it to learn generalised signals from the teacher. Moreover, by polluting the inputs (e.g., shuffling), the student is encouraged to capture salient phrases in addition to semantic matching, being more robust to possible perturbed inputs (Figure 4). After completing pre-training on synthetic queries, we further finetune the model on labelled data. Through iterating the pre-training and fine-tuning steps by using the latest model to relabel synthetic data, the resulting model can significantly outperform the one trained only using labelled data. In summary, the contributions are as follows:

1. We proposed a novel self-training framework to make use of automatically generated data more wisely for neural retrieval models (§3).
2. Our experimental results on general-domain benchmarks show that the proposed framework can not only significantly boost the performance of state-of-the-art neural retrievers, but also yield superior results in low-resource settings. We further show our framework is general and can be extended to improve more powerful cross-encoder-based rerankers (§4).
3. Further experiments are conducted on the out-of-domain BEIR benchmark (Thakur et al.,

2021), and both neural retrievers and rerankers surpass a series of strong models (§4.6).

## 2 Preliminaries

### 2.1 Task Definition

We focus on the task of short-passage retrieval (IR) in this work. Given a query in the form of a short text, the task requires retrieving a small set of passages that can satisfy the information needs, from a collection of passages in million or billion scale. Formally, suppose we have a passage collection $\mathcal{P} = \{p_1, p_2, \cdots, p_n\}$, the retriever is required to fetch top-$k$ passages $\mathcal{P}_q = \{p_1, p_2, \cdots, p_k\}$ from $\mathcal{P}$ that are relevant to a specific query $q$.

### 2.2 Dense Passage Retrieval

In contrast to traditional IR methods, such as BM25 (Robertson and Zaragoza, 2009), which represent texts in high dimensional and sparse vectors with inverted index, dense retrieval methods alternatively adopt neural models to encode texts (queries or passages) in dense latent vectors with much smaller dimensions. A dense passage retrieval model (Karpukhin et al., 2020) typically adopts the dual-encoder architecture, where neural models are used to encode the query and passage into dense vectors separately. The relevance is measured by the dot product between their embeddings:

$$s(q, p; \theta) = \mathbf{E}_q(q; \theta)^\top \cdot \mathbf{E}_p(p; \theta) \qquad (1)$$

where $\mathbf{E}_\cdot(\cdot; \theta)$ is an encoder parameterised by $\theta$. The adoption of this form of 'dual-encoder' architecture decouples the encoding of query and passage. At inference, all passages in $\mathcal{P}$ can be encoded offline. When a query $q$ comes in, efficient

nearest neighbour search (Johnson et al., 2021) can be performed to fetch the top-$k$ passages.

Contrastive learning is applied to train the dual-encoder. Given a query $q$, we have a positive passage $p^+$ and a set of $n$ negative passages $\mathcal{P}_q^- = \{p_i^-\}_{i=1}^n$. The model is being optimised by minimising the negative log likelihood of the positive passage:

$$
\begin{aligned}
\mathcal{L}_{\text{CL}} &= -\log D(p^+|q, \mathcal{P}, \theta) \\
&= -\log \frac{s(q, p^+; \theta)}{\sum_{p \in \{p^+\} \cup \mathcal{P}_q^-} s(q, p; \theta)} \quad (2)
\end{aligned}
$$

$\mathcal{P}_q^-$ is the set of irrelevant passages constructed from in-batch negatives (Chen et al., 2020) (i.e. positive passages of other queries in the same mini-batch) and mined hard negatives from existing retrievers (Karpukhin et al., 2020; Xiong et al., 2021).

## 3 Method

### 3.1 Self-Training with Synthetic Queries

Self-training (Yarowsky, 1995) has been shown effective in improving model performance through using unlabelled or synthetic data. It first use labelled data to train a good teacher, then the teacher is used to generate pseudo labels on unlabelled data. Finally, an identical student model is first pre-trained on unlabelled data with soft labels and then finetuned on labelled data. Recently, it has been shown to work well for a variety of tasks, including image classification (Xie et al., 2020) and neural machine translation (He et al., 2020). However, its effectiveness has not yet been evaluated for dense passage retrieval, especially with automatically synthesised queries.

Suppose we have a well-trained query generator, which is used to generate a set of synthetic query-passage pairs $\mathcal{T}' = \{(q', p'^+)\}$. Moreover, we assume that a teacher dense retrieval model that is fully trained on labelled data is available. The student retriever has the same architecture as the teacher, but it only accesses the soft labels produced by the teacher during training. This ensures that all involved negatives are not treated equally but with different soft labels, and more relevant passages are more likely to be penalised less. Moreover, we assume the use of teacher supervision will make the supervision signal for problematic queries (e.g., general or ambiguous queries) much more diverse, as the teacher will have little clue and a corresponding high entropy predictive distribution. Formally,

the learning process is conducted via KL divergence by closing the distribution distance between the student and teacher:

$$
\mathcal{L}_{\text{ST}} = \text{KL}(T(\cdot|q', \mathcal{P}'_{q'}, \theta_T), S(\cdot|q', \mathcal{P}'_{q'}, \theta_S)) \quad (3)
$$

where $\mathcal{P}'_{q'} = \{p'^+\} \cup \mathcal{P}'^-_{q'}$, which is the union of the pseudo positive passage of synthetic query $q'$ and the sampled negatives as in §2.2. $T(\cdot|q', \mathcal{P}'_{q'}, \theta_T)$ and $S(\cdot|q', \mathcal{P}'_{q'}, \theta_S)$ are the distributions from the teacher and student, respectively.

We further inject noises into student's inputs, hoping that it can generate more robust embeddings. Through injecting noises, the student needs to imitate the behaviour of the teacher with perturbed inputs (i.e. different views), being encouraged to learn generalised signals from the teacher (He et al., 2020). Inspired by Wu et al. (2019), we use following strategies for noise injection:

1. Shuffle: randomly choose some words in the query or passage as candidates for shuffling, then randomly shuffle these candidates.
2. Delete: randomly delete some words in the query or passage.
3. Mask: randomly mask some words in the query or passage with a [MASK] token.

Empirically, we found that applying them sequentially to both queries and passages with a probability 0.1 achieves the best results. Thus, the self-training loss of noised version becomes:

$$
\mathcal{L}_{\text{ST}} = \text{KL}(T(\cdot|q', \mathcal{P}'_{q'}, \theta_T), S(\cdot|\widetilde{q'}, \widetilde{\mathcal{P}}'_{q'}, \theta_S)) \quad (4)
$$

where $\widetilde{q'}$ and $\widetilde{\mathcal{P}}'_{q'}$ are noised query and passages.

### 3.2 Training Pipeline

Algorithm 1 summarises the training pipeline, with an overview shown in Figure 1. More specifically:

**Teacher Preparation**   (line 3 in Alg. 1) We first train a teacher model in labelled data using a two-stage training similar to (Gao and Callan, 2022). In the first stage, the retriever is trained with hard negatives sampled from a BM25 retriever. Then, the retriever trained in the first stage is used to discover hard negatives, which are later used to train a second-stage retriever. The resulted retriever serves as the teacher $\theta_T$ in our algorithm.

**Index Building & Hard Negative Mining**   (lines 4-5 in Alg. 1) The teacher model $\theta_T$ encodes all passages on $\mathcal{P}$ into dense vectors, which are later used to build the ANN search index with FAISS.

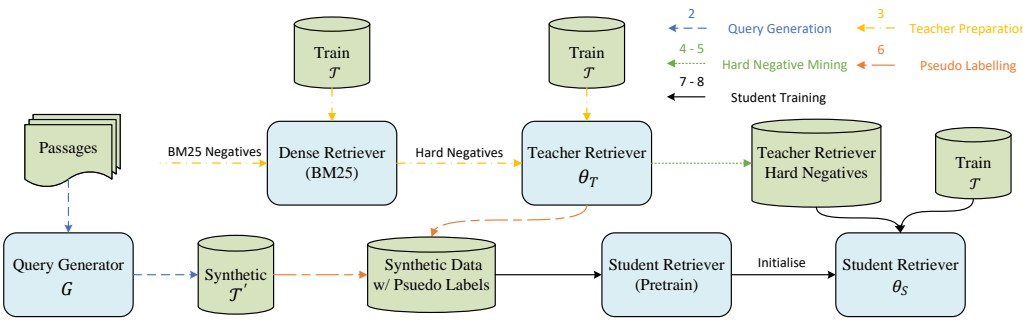

Figure 1: The overview of noisy self-training algorithm. Line numbers in Algorithm 1 are above each arrow type.

---

**Algorithm 1** Noisy Self-Training with Synthetic Queries

---

**Require:** Gold query-passage pairs $\mathcal{T} = \{(q, p^+)\}$
**Require:** Passage collection $\mathcal{P}$
1: Train a query generator $G$ on gold pairs $\mathcal{T}$.
2: Generate queries from $\mathcal{P}$ and construct synthetic query-passage pairs $\mathcal{T}' = \{(q', p'^+)\}_1^{\|\mathcal{P}\|}$.
3: Train a dual-encoder retriever $\theta_T$ on gold pairs $\mathcal{T}$.
4: Use $\theta_T$ to build ANN index on $\mathcal{P}$
5: Retrieve negatives $\mathcal{P}_q^-$ for each $(q, p^+) \in \mathcal{T}$ and $\mathcal{P}_{q'}^-$ for each $(q', p'^+) \in \mathcal{T}'$
6: Use $\theta_T$ as the teacher to generate soft-labels on synthetic queries $\mathcal{T}' = \{(q', p'^+, \mathcal{P}_{q'}^-)\}$
7: Pre-train a student retriever $\theta_S$ on soft-labelled synthetic queries $\mathcal{T}'$, with noises injected to $\theta_S$'s inputs
8: Finetune $\theta_S$ on gold pairs $\mathcal{T} = \{(q, p^+, \mathcal{P}_q^-)\}$
9: Take $\theta_S$ as new teacher: $\theta_T \leftarrow \theta_S$ and go back to **line** 6

---

Hard negatives for both labelled data $\{(q, p^+)\}$ and synthetic data $\{(q', p'^+)\}$ are retrieved using the built index, by taking $k$-nearest neighbours while excluding the gold passage.

**Noisy Self Training & Fine-tuning** (lines 6-8 in Alg. 1) A student model $\theta_S$ is first pre-trained on the synthetic queries with the noised self-training objective (Eq. 4), after which it is fine-tuned on labelled data according to Eq. 2.

**Iterative Training** (line 9 in Alg. 1) The teacher model can be replaced by the student to generate new pseudo labels and do noisy self-training and fine-tuning all over again.

## 4 General Domain Experiments

### 4.1 Datasets

We evaluate our method on two general-domain datasets: MS-MARCO passage ranking (Bajaj et al., 2016) and Natural Questions (NQ) (Kwiatkowski et al., 2019). MRR@10, Recall@50, Recall@1000 on the MS-MARCO dev set are reported, where MRR represents the Mean Reciprocal Rank, which is calculated as the sum of the reciprocal rank of the first retrieved relevant passage for each query; and Recall@k is the proportion of relevant passages that appear in the top-k retrievals. Recall@k (k=5, 20, 100) is reported on Natural Questions, which is the proportion of top-k retrieved passages that contain the answer string to each query. The evaluation script provided by Pyserini (Lin et al., 2021) is used for all experiments.

### 4.2 Experimental Settings

We replicate the coCondenser model (Gao and Callan, 2022) using PyTorch (Paszke et al., 2019) and treat it as our baseline.

For self-training, the coCondenser model that was fully trained on labelled data serves as the teacher, which we use to train the student retriever on synthetic queries. After self-training, the student is further finetuned on labelled data, with hyperparameters following Gao and Callan (2022). The last checkpoint is selected for evaluation on test set for all experiments. More details are in Appendix A.1.

### 4.3 Main Results

Table 2 shows the results of our model compared with a range of lexical and neural retrievers. We observe that the performance of our replicated coCondenser is competitive with that of Gao and Callan (2022), achieving slightly better results on Natural Questions but slightly worse on MS-MARCO. By applying our proposed noisy self-training framework, the student coCondenser outperforms the state-of-the-art results on both datasets, resulting in significant improvements over coCondenser (1.2% MRR@10 and R@5 on MS-MARCO and Natural Questions, respectively). This also shows that although coCondenser has already been pre-trained on the target corpus, continuing pre-training it on synthetic queries with noisy self-training can still lead to further performance boost.

To test if the proposed method is general, we also combine it with various pre-trained models, includ-

| Method | MS-MARCO | | | Natural Questions | | |
|---|---|---|---|---|---|---|
| | MRR@10 | R@50 | R@1k | R@5 | R@20 | R@100 |
| BM25 (Robertson and Zaragoza, 2009)* | 18.7 | 59.2 | 85.7 | - | 59.1 | |
| DeepCT (Dai and Callan, 2020)* | 24.3 | 69.0 | 91.0 | - | - | - |
| docT5query (Nogueira and Lin, 2019)* | 27.7 | 75.6 | 94.7 | - | - | - |
| GAR (Mao et al., 2021)* | - | - | - | 60.9 | 74.4 | 85.3 |
| DPR (Karpukhin et al., 2020)* | - | - | - | - | 74.4 | 85.3 |
| ANCE (Xiong et al., 2021)* | 33.0 | - | 95.9 | - | 81.9 | 87.5 |
| ColBERT (Khattab and Zaharia, 2020)* | 36.0 | 82.9 | 96.8 | - | - | - |
| DPR-PAQ (Oguz et al., 2022)* | | | | | | |
| - BERT$_{base}$ | 31.4 | - | - | 74.5 | 83.7 | 88.6 |
| - RoBERTa$_{base}$ | 32.3 | - | - | 74.2 | 84.0 | 89.2 |
| Condenser (Gao and Callan, 2021)* | 36.6 | - | 97.4 | - | 83.2 | 88.4 |
| coCondenser (Gao and Callan, 2022)* | 38.2 | - | 98.4 | 75.8 | 84.3 | 89.0 |
| Our coCondenser (Teacher) (Gao and Callan, 2022)[†] | 37.8 | 85.5 | 98.2 | 75.9 | 85.0 | 89.4 |
| Student coCondenser | **39.4** | **86.7** | **98.5** | **77.0** | **85.5** | **89.5** |

Table 2: Results on MS-MARCO dev set and Natural Questions test set. ∗ indicates results directly copied from Gao and Callan (2022) and Khattab and Zaharia (2020). † indicates our implementation. The best results are marked bold and unavailable results are left blank.

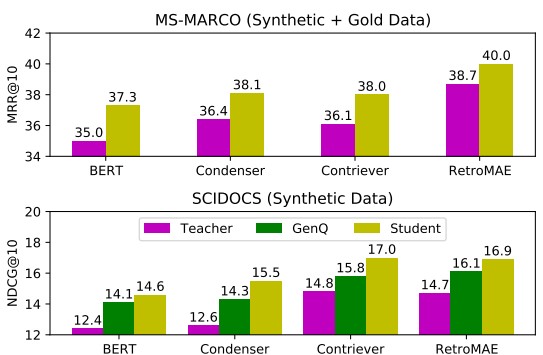

Figure 2: Results on MS-MARCO dev and SCIDOCS test sets when using different pre-trained models.

| Method | MS-MARCO | Natural Questions | | |
|---|---|---|---|---|
| | MRR@10 | R@1 | R@5 | R@20 |
| Teacher Reranker | 43.0 | 61.9 | 80.0 | 86.4 |
| Student Reranker | **43.7** | **62.7** | **80.3** | **86.8** |

Table 3: Compared results between the teacher and student reranker. The top-50 and top-100 predictions from student coCondenser are re-ranked on MS-MARCO and Natural Questions, respectively.

ing BERT (Devlin et al., 2019), Contriever (Izacard et al., 2021), Condenser (Gao and Callan, 2021), and RetroMAE (Xiao et al., 2022). Note that all models share the same architecture and only differ in their initial weights. Figure 2 shows the results on MS-MARCO. It is observed that the student model achieves consistent improvements on all pre-trained models. Moreover, for worse teachers, the benefits gained by adopting the proposed method are generally more significant (e.g., 2.3 MRR@10 on BERT vs 1.3 MRR@10 on RetroMAE).

| Method | MS-MARCO | | NQ | |
|---|---|---|---|---|
| | MRR@10 | R@50 | R@5 | R@20 |
| Student coCondenser | **39.4** | **86.7** | **77.0** | **85.5** |
| w/o noises | 38.8 | 86.6 | 76.7 | **85.5** |
| w/o pseudo labels | 38.8 | 84.5 | 75.9 | 84.2 |
| consistency filtering | 38.8 | 86.2 | 76.5 | 85.3 |
| joint training | 38.4 | 86.1 | 76.8 | 85.3 |

Table 4: Results of different variants compared to the student model on MS-MARCO and Natural Questions.

We also adapt the proposed framework to see if it can be applied to improve an expensive reranker. Similarly, a teacher reranker that was fully trained on labelled data[2] is used to generate soft labels on synthetic data, and an identical student reranker is pre-trained. A cross-encoder model is used as the backbone to jointly encode both query and passage. As shown in Table 3, we see that by reranking the student coCondenser's top-$k$ predictions, the student reranker can outperform the teacher, boosting the performance with another 0.7% and 0.8% gains on MRR@10 and R@1, respectively.

### 4.4 Ablation Studies

We conduct ablation studies to further understand our methods, with results reported in Table 4.

**Noise Injection** We remove noise injection during pre-training, and the pre-training objective changes from Eq. 4 to Eq. 3. It is observed that the noisy injection strategies show positive impacts in helping the model rank correct answers higher.

---

[2]The teacher retriever hard negatives in Figure 1 are used as negative examples for training. See Figure 5 for full details.

**Pseudo Labels** We remove **line** 4 in Algorithm 1 and directly use synthetic labels $\{(q', p'^+)\}$ together with hard negatives for pre-training in **line** 5. The results show that directly pre-training on synthetic query-passage pairs leads to inferior performance on both datasets, resulting in significant performance degradation on all metrics, especially on Natural Questions where the R@20 score is even worse than the coCondenser baseline. This indicates that synthetic query-passage pairs contain a large number of noises, and adopting our proposed method can effectively alleviate the negative impacts of noises, which validates our hypothesis.

**Consistency Filtering** We take the teacher as a consistency filter (Alberti et al., 2019) to remove noises contained in synthetic data. More specifically, for a given synthetic query-passage pair $(q', p'^+)$, if $p'^+$ can be retrieved by the teacher in the top-1 position, this pair is kept; otherwise, it will be discarded. Although this strategy can effectively improve the quality of synthetic data and also yields competitive performance, taking the teacher as a pseudo-label generator leads to better results.

**Joint Training** We jointly train the student model on both synthetic and labelled data, where the loss becomes $\mathcal{L} = \mathcal{L}_{ST} + \mathcal{L}_{CL}$. Different batch sizes are also used for synthetic and labelled data to ensure the overall update steps are roughly the same as in pre-train + finetune. We observe that joint training leads to significantly worse results on MS-MARCO but comparable performance on Natural Questions.

### 4.5 Analysis

**Data Efficiency** We further analyse how the size of labelled query-passage pairs used for training affects the retrieval performance of our method. Smaller datasets are randomly sampled from the full MS-MARCO training data with different sizes, ranging from 1% to 70%. Note that in each data size setting, all involved models are restricted to that specific amount of labelled data samples.[3] Figure 3 compares the performance of the teacher and student model trained on labelled data with different sizes. The results reveal that the student surpasses the teacher under all data sizes, and the performance gap becomes larger as the number of

---

[3]The teacher retriever, query generator and student retriever are all trained on the same amount of labelled data in each setting. The size of synthetic data remains unchanged across all settings.

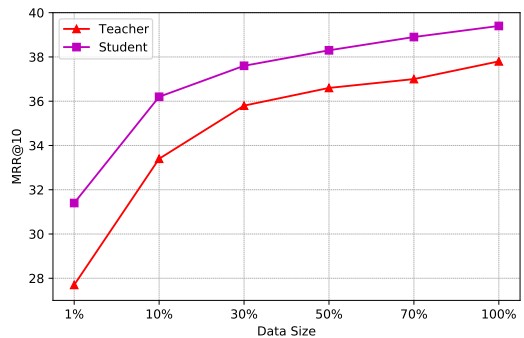

Figure 3: Impacts of labelled training data size on MS-MARCO dev set. Teacher refers to coCondenser.

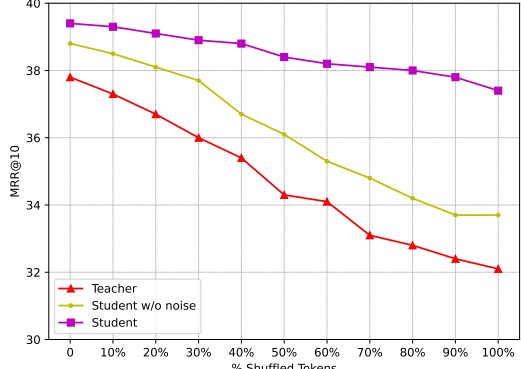

Figure 4: Impacts of randomly shuffling different proportions of tokens on MS-MARCO dev queries at test time. Teacher refers to coCondenser.

labelled data samples decreases. Moreover, the student is superior in terms of data efficiency, achieving performance comparable to the teacher model using full data samples (i.e., our coCondenser in Table 2) with as little as 30% of labelled data, and the performance continues to improve when more data is available.

**Robustness to Shuffled Query** Figure 4 compares the performance when different proportions of tokens in each test query are randomly shuffled. We observe that the student model is significantly more robust to queries with tokens in random orders. By removing noise injection during training, the performance drops become increasingly larger with more tokens shuffled, indicating that the noise injection strategy not only leads to performance gains in normal settings but also improves the robustness towards perturbed inputs. This also signifies our model learns lexical matching in capturing salient phrases to some extent, a property typically held in sparse retrievers (e.g., BM25).

### 4.6 Out-of-Domain Experiments

We further test the domain adaptation ability of our proposed method by reporting the performance on the BEIR benchmark (Thakur et al., 2021), which

| Model ($\rightarrow$) | Lexical & Sparse | | | Dense Retriever | | | | | | Retriever + Reranker | | |
|---|---|---|---|---|---|---|---|---|---|---|---|---|
| | BM25* | DeepCT* | docT5query*§ | Teacher | GenQ†§ | GPL¶§ | GPL-S | PTR§ | Student Retriever§ | PTR++§ | Teacher Reranker | Student Reranker§ |
| **Model Size** | - | - | - | 110M | 110M×18 | 66M×18 | 110M | 110M×11 | 110M | 110M×11 | 110M | 110M |
| **QGen. Size** | - | - | 220M | - | 220M | 220M | 220M | 137B | 220M | 137B | - | 220M |
| TREC-COVID | 65.6 | 40.6 | 71.3 | 74.1 | 68.4 | 70.0 | 73.6 | 72.7 | **76.7** | 76.0 | **77.6** | 76.5 |
| BioASQ | **46.5** | 40.7 | 43.1 | 29.5 | 31.0 | 44.2 | 44.7 | - | 35.7 | - | **46.8** | **46.8** |
| NFCorpus | 32.5 | 28.3 | 32.8 | 33.1 | 34.6 | 34.5 | **34.9** | 33.4 | **34.9** | 36.0 | 31.0 | 32.6 |
| NQ | 32.9 | 18.8 | 39.9 | 48.4 | 44.4 | 48.3 | 49.0 | - | **50.3** | - | 58.5 | **59.1** |
| HotpotQA | 60.3 | 50.3 | 58.0 | 56.9 | 56.5 | 58.2 | 56.3 | **60.4** | 58.4 | **71.2** | 66.6 | 68.4 |
| FiQA-2018 | 23.6 | 19.1 | 29.1 | 30.1 | 33.0 | 34.4 | 34.6 | **40.4** | 33.3 | **45.9** | 39.1 | 39.6 |
| Signal-1M (RT) | **33.0** | 26.9 | 30.7 | 27.8 | 27.0 | 27.6 | 27.0 | - | 29.5 | - | 26.5 | **28.1** |
| TREC-NEWS | 39.8 | 22.0 | 42.0 | 40.1 | 40.7 | 42.1 | 42.8 | - | **43.8** | - | 43.9 | **44.2** |
| Robust04 | 40.8 | 28.7 | 43.7 | 43.6 | 40.6 | 43.7 | **46.9** | - | 46.8 | - | 46.2 | **48.6** |
| ArguAna | 31.5 | 30.9 | 34.9 | 37.3 | 46.3 | **55.7** | 53.4 | 53.8 | 50.1 | 52.1 | 55.1 | **57.8** |
| Touché-2020 | **36.7** | 15.6 | 34.7 | 30.5 | 17.9 | 25.5 | 24.4 | 26.6 | 30.4 | 27.8 | 33.3 | **35.0** |
| CQADupStack | 29.9 | 26.8 | 32.5 | 32.6 | 35.5 | 35.7 | **36.2** | - | 35.8 | - | 35.0 | 35.5 |
| Quora | 78.9 | 69.1 | 80.2 | 85.6 | 86.1 | 83.6 | 83.6 | - | **86.8** | - | 80.0 | 82.5 |
| DBPedia | 31.3 | 17.1 | 33.1 | 37.5 | 36.3 | **38.4** | 38.1 | 36.4 | 37.2 | 41.3 | 41.5 | **43.3** |
| SCIDOCS | 15.8 | 12.4 | 16.2 | 14.3 | 15.2 | **16.9** | 16.7 | 16.3 | 16.1 | **19.1** | 15.6 | 16.4 |
| FEVER | 75.3 | 35.3 | 71.4 | 71.8 | 69.2 | 75.9 | 75.9 | **76.2** | 72.2 | **83.8** | 82.1 | 82.1 |
| Climate-FEVER | 21.3 | 6.6 | 20.1 | 17.6 | 21.8 | **23.5** | 23.4 | 21.4 | 22.2 | 22.6 | **23.3** | **23.3** |
| SciFact | 66.5 | 63.0 | 67.5 | 61.0 | 66.2 | 67.4 | **67.6** | 62.3 | 65.1 | **73.2** | 70.8 | 72.1 |
| Avg. Performance | | | | | | | | | | | | |
| PTR-11 Subsets | 41.8 | 29.0 | 42.6 | 42.2 | 42.3 | **45.5** | 45.4 | **45.5** | 45.2 | **49.9** | 48.7 | 49.7 |
| All | 42.3 | 30.7 | 43.4 | 42.9 | 42.8 | 45.9 | **46.1** | - | 45.9 | - | 48.5 | **49.6** |

Table 5: Results on BEIR benchmark (nDCG@10). Best results are marked bold and second best results are underlined. ∗ indicates results copied from Thakur et al. (2021). † indicates our implementation. § means methods using synthetic queries. ¶ means methods learning from cross-encoder rerankers, thus are not directly comparable to ours. ×$n$ means that these methods train specialised models for each datasets.

contains 18 datasets from retrieval tasks of diverse formats and domains. The average nDCG@10 score over all datasets is used for evaluation.

**Implementation Details** We use the coCondenser model that was fine-tuned on MS-MARCO as the teacher model in our algorithm, which we call **Teacher** henceforth. For synthetic queries, we directly use the publicly available ones released by Wang et al. (2022).[4] Since labelled data is not available, the fine-tuning step is eliminated. Unlike previous methods that train specialised retrievers for each task, we train a single *universal* retriever on the union of synthetic queries from all tasks. We directly use the last checkpoint for evaluation on the test set of each task. To ensure fair comparison, we also re-implement GPL by following the same configurations as in Wang et al. (2022) except that

we employ the **Teacher** retriever for initialisation[5] and train a single model, denoted as **GPL-S**.

We also investigate if the proposed method can improve the reranker's out-of-domain performance. Similarly, the reranker trained on MS-MARCO serves as the teacher: **Teacher-Reranker**, guided by which an indentical **Student Reranker** is trained. At inference, both rerankers are applied to rerank the top-100 predictions of the **Student Retriever**. The official evaluation script is used for experiments.[6] More details are in Appendix A.2.

**Main Results** We compare our proposed model with a variety of models, including **BM25** (Robertson and Zaragoza, 2009), **DocT5query** (Nogueira and Lin, 2019), **DeepCT** (Dai and Callan, 2020), **GenQ** (Thakur et al., 2021), **GPL** (Wang et al.,

---

[4] https://public.ukp.informatik.tu-darmstadt.de/kwang/gpl/generated-data/beir

[5] It achieves better performance compared to TAS-B used in their original implementations.

[6] https://github.com/beir-cellar/beir

2022) and **PTR** (Dai et al., 2023).[7] Table 5 shows the experimental results. We first notice that the **Teacher** model already achieves strong results compared to other baseline models. GenQ which further finetunes **Teacher** on synthetic data does not show positive effects, indicating that synthetic queries are extremely noisy and directly using them for training is not beneficial. By contrast, the student retriever trained using our noisy self-training framework significantly improves over **Teacher**, increasing the averaged nDCG@10 by 3.0%. Even when being compared to GPL and GPL-S, the models that also use query generation for training data augmentation but take a prohibitively expensive reranker as a pseudo-label generator, and PTR that prompts a 137B instruction-tuned large language model for query generation, our model can still achieve competitive averaged performance and beats these task-specialised retrievers on 6 out of 18 tasks while being comparable on the rest. Note that our model does not rely on any external models, and it is improved in a self-evolution manner, managing to exhibit a high degree of efficiency. More specifically, when being compared to GPL, our method significantly diminishes training time by roughly a factor of three (204h → 74h), and it achieves a 25× speed-up in relevance labelling (20ms/q → 0.8ms/q). Figure 2 shows that our method achieves consistent improvements and beats GenQ across different pre-trained models on SCIDOCS, a dataset that is significantly different from MS-MARCO in retrieval needs and domain. See Table 10 for full details.

When adopting the same framework to rerankers, the student rereanker can further boost the performance with another 1.1 average points over the teacher reranker. The results again confirm that our method is general and can be extended to improve a powerful reranker in out-of-domain settings.

**Discussion** The adaptation of our proposed framework in out-of-domain tasks can be interpreted as a specific type of unsupervised domain adaptation (UDA) algorithm (Wang and Deng, 2018), where a model aims to maximise its performance on target domains when only labelled data from in-domain sources and unlabelled data from target domains are available. In our case, the query generator trained on in-domain data (i.e., MS-MARCO) is employed to generate synthetic queries on unlabelled target-domain data (i.e., cor-

[7]See Appendix B for more baseline details.

pus in BEIR datasets). Meanwhile, the retriever trained on in-domain data (i.e. **Teacher**) generates soft labels for these synthetic queries. As a result, the query generator fabricates distributions of potential queries that could be possibly asked in the target domain; while the teacher retriever captures prevalent patterns of correspondence between queries and passages within the target domain, by leveraging the knowledge it has acquired from the labelled in-domain data. When exposing it to such silver target-domain data during training, a new student retriever is enforced to acquire relevant knowledge that is required to complete retrieval tasks in target domains. Consequently, the exposure to pseudo target-domain data mandates the improvement of domain-specific aptitude within the student retriever, allowing it to effectively retrieve information within target domains.

### 4.7 Self-Training Iterations

Table 6 compares the models trained with varying numbers of iterations. We observe that employing a single iteration yields optimal results, while performance diminishes with more iterations. We conjecture that errors produced in relevance labelling may accumulate over successive iterations, potentially reinforcing the model's bias towards particular error types as the training process continues.

### 4.8 Quality of Pseudo Labels

To assess the quality of the generated soft labels, we randomly sampled 100 synthetic queries and verified the relatedness of the originating passage to each synthetic query through manual examination. We observed that 49 of these queries were indeed related to their source passages. Among this subset of 49 queries, we noted that, 93% of the time, the soft labels effectively identified alternative passages capable of accurately responding to the query. This was indicated by the assignment of a high probability mass to these alternative passages. In contrast, for the remaining 51 query-passage pairs, where the association was found to be incorrect, soft labels can identify better-matched passages for 42 of them. This observation provides strong evidence for the effectiveness of our proposed self-training methodology.

## 5 Related Work

**Neural Dense Retriever** Neural retrievers adopt neural networks to encode texts into low-

| Iteration | MM | TC | BA | NF | NQ | HP | FQ | SG | TN | RB | AA | T2 | CQ | QU | DB | SD | FE | CF | SF | Avg |
|---|---|---|---|---|---|---|---|---|---|---|---|---|---|---|---|---|---|---|---|---|
| 0 | 37.3 | 74.1 | 29.5 | 33.1 | 48.4 | 56.9 | 30.1 | 27.8 | 40.1 | 43.6 | 37.3 | **30.5** | 32.6 | 85.6 | **37.5** | 14.3 | 71.8 | 17.6 | 61.0 | 42.9 |
| 1 | **39.4** | **76.7** | 35.7 | **34.9** | **50.3** | **58.4** | 33.3 | **29.5** | **43.8** | **46.8** | 50.1 | 30.4 | 35.8 | **86.8** | 37.2 | 16.1 | **72.2** | **22.2** | 65.1 | **45.9** |
| 2 | 39.1 | 75.7 | **36.9** | **34.9** | 49.2 | 57.5 | **33.7** | 28.7 | 42.8 | 46.4 | **50.6** | 29.2 | **36.5** | 86.2 | 35.8 | **16.5** | 70.9 | 22.2 | **66.7** | 45.6 |

Table 6: Ablations on the number of self-training iterations. Tasks are ordered as in Table 5. MM indicates MS-MARCO and is excluded when computing the average.

dimensional embeddings and show superiority over lexical retrievers when being trained on sufficient data. Karpukhin et al. (2020) adopts a dual-encoder structure to use two independent neural encoders to encode queries and passages into fix-sized vectors separately with their dot product as the relevance score. The model is trained to discriminate positive passages from randomly-sampled irrelevant ones or more informative negatives (Xiong et al., 2021). Other works adopt the poly-encoder architecture (Humeau et al., 2020), where query and documents are represented as multiple vectors to allow token-level interactions (Khattab and Zaharia, 2020). Although effective, the benefits come at the cost of increased index size and a more complex scoring function. In this work, we adopt the dual-encoder structure for its simplicity. However, our proposed method is orthogonal to model architectures, and we believe its combination with poly-encoder retrievers is worth further exploration.

**Synthetic Queries for Information Retrieval** Synthetic queries have been widely used in information retrieval. Early work expands passages with synthetic queries for lexical retrievers (Nogueira and Lin, 2019). Recent neural models take synthetic queries and their originating passages as positive pairs for model pre-training, resulting in boosted performance (Lu et al., 2021), better domain adaptation ability (Ma et al., 2021; Gangi Reddy et al., 2022) and promising zero-shot results (Dai et al., 2023). In order to reduce the noise in synthetic data, Wang et al. (2022) exploits a cross-encoder reranker to generate pseudo labels. It incurs expensive costs in generating soft labels and training task-customised retrievers, resulting in slow adaptation. Our work follows this direction but significantly differs in that we do not rely on any external model for synthetic data labelling and create dataset-specialised retrievers. By contrast, we empirically show that by using the retriever itself as a more efficient pseudo-label generator, it can be improved in a self-evolution manner with our introduced noisy self-training framework. Moreover, we show this framework is general and can

be extended to boost reranking performance.

**Self-Training** Self-training (Scudder, 1965) refers to a class of approaches that learn from unlabelled data with pseudo labels. A good teacher model is first trained on labelled data, which is later used to label unlabelled data. Another student model is then pre-trained on unlabelled data first and further finetuned on labelled data. More advanced methods train multiple teachers using features of disjoint partitions on labelled data and a student is learned from their ensembles (Blum and Mitchell, 1998). Recently, the effectiveness of self-training has been verified in a wide range of tasks, including machine translation with back translation (Wu et al., 2019) and language generation (He et al., 2020). These approaches are termed noisy self-training, as the inputs of the student are perturbed. In this work, we follow this direction to show that, for the first time, noisy self-training can be adopted to make better use of synthetic queries to improve neural retrievers in both general-domain and out-of-domain settings.

## 6 Conclusion

In this paper, we present a novel noisy self-training framework for neural retrieval models. It shows that when combined with automatically generated queries, neural retrievers can be improved in a self-evolution manner without relying on any external models. Empirical results on both general-domain and out-of-domain benchmarks confirm the superiority of our proposed method, significantly outperforming a wide range of competitive existing models. We further adapt our method to show it can be applied to improve an expensive reranker.

## Limitations

Although our proposed method does not change model architectures and the resulting models can perform as efficiently as previous models, the introduced training framework does incur the additional training cost, including query generation and the associated hard negative mining. Despite these ex-

tra costs, the training of our methods has a fairly modest footprint by modern standards, taking about 3 days of a server with 4×A100 GPUs and 450G CPU RAM.

We mainly experiment with dual-encoder-based neural dense retrieval models and additionally adapt the method to reranker training in this work. Alternative retriever methods such as ColBERT (Khattab and Zaharia, 2020) and SPLADE (Formal et al., 2021) are compatible with our approach, and their incorporation should lead to further gains. However, they introduce extra complexity compared to the dual-encoder, e.g., requiring an index over tokens rather than complete passages. We leave their efficient integration as future work.

For out-of-domain experiments, the current method still relies on using general-domain datasets to obtain the teacher model and the query generator. How to achieve this in an unsupervised setting needs further exploration. Moreover, on tasks that have different retrieval needs from general passage retrieval, synthetic queries are normally quite different from the gold standard. For instance, some gold queries in DBPedia are general and may have multiple matching passages (e.g., *Give me all people that were born in Vienna and died in Berlin.*), but synthetic queries are usually only related to their originating passages (e.g., *Who was Johannes Mayer*). How to generate synthetic queries with high quality and similar properties to gold standard queries is the key to better retrievers.

As this work uses pre-trained language models for generating synthetic queries, it is possible that undesirable biases (e.g., gender and cultural) from the language models (Wei et al., 2022) is propagated to downstream models. This is in additional to existing biases in training and evaluation datasets (Bigdeli et al., 2021). Evaluating the extent to which biases affect the synthetic data and the resulting model is an inherently complex problem, and remains an open question for future work.

## Acknowledgements

We thank the anonymous reviewers for their helpful feedback and suggestions. The first author is supported by the Graduate Research Scholarships funded by the University of Melbourne. This work was funded by the Australian Research Council, Discovery grant DP230102775. This research was undertaken using the LIEF HPCGPGPU Facility hosted at the University of Melbourne, which was established with the assistance of LIEF Grant LE170100200.

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

## A Implementation Details

Statistics about the datasets used in our experiments are reported in Table 7. Tables 8 and 9 show the hyperparameters used in our experiments.

### A.1 General Domain Datasets

`co-condenser-marco`[8] is used for model initialisation on MS-MARCO and `co-condenser-wiki`[9] on Natural Questions. For retrievers, the weights of dual encoders are tied. The teacher is trained using the two-stage training following the hyperparameters used in (Gao and Callan, 2022) and (Karpukhin et al., 2020). The student is pre-trained for 1 epoch for all datasets, with a learning rate $1 \times 10^{-5}$ and batch size 32. The top-100 passages retrieved by the teacher are used as hard negatives, from which 7 negatives are sampled for each query in a mini-batch. We empirically find conducting the training with one iteration is enough for achieving the best result. The training takes about 48 hours on both datasets, with up to 4 A100 GPUs.

For generating synthetic queries, we use the publicly available model[10] to generate queries on MS-MARCO with top-k sampling. For Natural Questions, we first finetune a T5 model on labelled query-passage pairs for 200 epochs with learning rate $2 \times 10^{-5}$, which takes about 10 hours on 1 A100 GPU. One query per passage is generated for all datasets, which requires approximately 7 hours on MS-MARCO and 16 hours on Natural Questions using 2 A100 GPUs.

As for the reranker described in §3, ERNIE-base[11] is used for model initialisation. The teacher reranker is trained with the teacher retriever hard negatives (Figure 1). It is trained for 2 epochs on MS-MARCO, with learning rate set to $1 \times 10^{-5}$, batch size 12, and weight decay 0.1. Each query is paired with 40 sampled negatives; while for Natural Questions, we train it for 10 epochs with 15 sampled negatives. For pre-training the student reranker, the batch size is set to 48 and 128 on MS-MARCO and Natural Questions, respectively, and other hyperparameters remain the same. For the finetuning stage, the settings used for training

---

[8]https://huggingface.co/Luyu/co-condenser-marco
[9]https://huggingface.co/Luyu/co-condenser-wiki
[10]https://huggingface.co/BeIR/query-gen-msmarco-t5-base-v1
[11]https://huggingface.co/nghuyong/ernie-2.0-base-en

---

the teacher reranker are adopted. Completing the whole training pipeline requires approximately 80 hours on MS-MARCO and 67 hours on Natural Questions with up to 4 A100 GPUs.

### A.2 Out-of-Domain Datasets

On BEIR benchmark, the **Teacher** retriever is used to mine hard negatives for synthetic queries on each dataset. The top-100 passages retrieved by the **Teacher** retriever are regarded as the hard negative pool. **Student Retriever** is initialised from the `co-condenser-marco` checkpoint. On each dataset, we train a **Student Retriever** for 10 epochs with learning rate $1 \times 10^{-5}$. The batch size is set to 32, and each query is paired with 7 sampled negatives. Training a single model on each dataset takes about 8 hours using 2 A100 GPUs and two iterations are used to achieve the best average performance. For GenQ, we follow Thakur et al. (2021) to further finetune the **Teacher** retriever on synthetic queries of each dataset for 1 epoch with batch size 64 and only in-batch negatives.

The reranker is initialised from ERNIE-base and finetuned for 5 epochs with learning rate $1 \times 10^{-5}$ and batch size 12. Each query is paired with 15 negatives sampled from the same hard negative pool as above. Training a single model takes about 18 hours using 2 A100 GPUs.

## B Baselines on BEIR

**BM25** (Robertson and Zaragoza, 2009) is a lexical retriever based on token matching. **DocT5query** (Nogueira and Lin, 2019) uses a query generator to synthesise queries and append them to passages as expansion. **DeepCT** (Dai and Callan, 2020) uses BERT model trained on MS-MARCO to compute the weight of each term in the vocabulary and each passage is represented with keywords multiplied by the term weights. **GenQ** (Thakur et al., 2021) first trains a dense retrieval model on MS-MARCO and continues to finetune it on synthetic queries with in-batch negatives through contrastive learning. **GPL** (Wang et al., 2022) uses knowledge distillation to train a dense retriever by learning from a reranker trained on MS-MARCO. **PTR** (Dai et al., 2023) generates a large number of synthetic queries by prompting an instruction-tuned large language model and trains task-specialised retrievers.

| Task | Domain | Dataset | Train | dev | test | #Passages |
|------|--------|---------|-------|-----|------|-----------|
| Passage Retrieval | Misc. | MS-MARCO | 502,939 | 6,980 | - | 8,841,823 |
| Open Domain QA | Wikipedia | Natural Questions | 58,880 | 8,757 | 3,610 | 21,015,324 |
| **BEIR Benchmark** | | | | | | |
| Biomedical | Biomedical | TREC-COVID | - | - | 50 | 171,332 |
| Information | Biomedical | BioASQ | - | - | 500 | 14,914,602 |
| Retrieval | Biomedical | NFCorpus | - | - | 323 | 3,633 |
| | Wikipedia | NQ | - | - | 3,452 | 2,681,468 |
| Question Answering | Wikipedia | HotpotQA | - | - | 7,405 | 5,233,329 |
| | Financial | FiQA-2018 | - | - | 648 | 57,638 |
| Twitter Retrieval | Twitter | Signal-1M (RT) | - | - | 97 | 2,866,316 |
| News Retrieval | News | TREC-NEWS | - | - | 57 | 594,977 |
| | News | Robust04 | - | - | 249 | 528,155 |
| Argument Retrieval | Misc. | ArguAna | - | - | 1,406 | 8,674 |
| | Misc. | Touché-2020 | - | - | 49 | 382,545 |
| Duplicate Question | StackEchange | CQADupStack | - | - | 13,145 | 457,199 |
| Retrieval | Quora | Quora | - | - | 10,000 | 522,931 |
| Entity Retrieval | Wikipedia | DBPedia | - | - | 400 | 4,635,922 |
| Citation Prediction | Scientific | SCIDOCS | - | - | 1,000 | 25,657 |
| | Wikipedia | FEVER | - | - | 6,666 | 5,416,568 |
| Fact Checking | Wikipedia | Climate-FEVER | - | - | 1,535 | 5,416,593 |
| | Scientific | SciFact | - | - | 300 | 5,183 |

Table 7: Number of query examples in Train/Dev/Test sets and passage number in each target corpus. - means unavailable data that was not used in our experiments. Datasets in BEIR benchmark are grouped according to Thakur et al. (2021), please refer to their paper for full details. TREC-COVID (Voorhees et al., 2021), BioASQ (Tsatsaronis et al., 2015), NFCorpus (Boteva et al., 2016), NQ (Kwiatkowski et al., 2019), HotpotQA (Yang et al., 2018), FiQA-2018 (Maia et al., 2018), Signal-1M (Suarez et al., 2018), TREC-NEWS (Soboroff et al., 2019), Robust04 (Voorhees, 2005), ArguAna (Wachsmuth et al., 2018), Touché (Bondarenko et al., 2020), CQADupStack (Hoogeveen et al., 2015), DBPedia (Hasibi et al., 2017), SCIDOCS (Cohan et al., 2020), FEVER (Thorne et al., 2018), Climate-FEVER (Leippold and Diggelmann, 2020), SciFact (Wadden et al., 2020).

## C  Adopting Different Pre-trained Models on BEIR

We test our method on the BEIR benchmark with different pre-trained models. As Table 10 shows, the student model trained using our method achieves consistent improvements on all kinds of pre-trained models, confirming the proposed algorithm is general enough to be adapted to diverse models on out-of-domain settings. Moreover, the patterns for improvements are similar to those in MS-MARCO, with more obvious gains on less-performing teachers (e.g., +4.0% on Contriever vs +2.6% on RetroMAE). GenQ shows positive effects on less-performing pre-trained models (i.e., BERT, Condenser, and Contriever). However, it struggles with beating the teacher retriever on more advanced pre-trained models and even degrades average performance. Besides, our method can consistently surpass GenQ across all settings, again confirming its superiority.

## D  License

1. MS-MARCO is licensed under the **Creative Commons Attribution 4.0 International**.
2. Natural Questions is licensed under the **Apache License 2.0**.
3. The BEIR benchmark is licensed under the **Apache License 2.0**.

| | Hyperparameters | MS-MARCO | Natural Questions | BEIR |
|---|---|---|---|---|
| **Shared** | Max query length | 32 | 32 | 32 |
| | Max Passage length | 128 | 128 | 128 |
| | Optimizer | AdamW | AdamW | AdamW |
| | Scheduler | Linear warmup | Linear warmup | Linear Warmup |
| | Warmup proportion | 0.1 | 0.1 | 0.1 |
| | Weight Tying | Yes | Yes | Yes |
| | #Params | 110M | 110M | 110M |
| **Teacher Preparation** | | | | |
| **Teacher Stage 1** | Learning rate | $5 \times 10^{-6}$ | $1 \times 10^{-5}$ | - |
| | Batch size | 8 | 128 | - |
| | #hard negatives | 7 | 1 | - |
| | Hard negative source | BM25 | BM25 | - |
| | #Epochs | 3 | 40 | - |
| **Teacher Stage 2** | Learning rate | $5 \times 10^{-6}$ | $1 \times 10^{-5}$ | |
| | Batch size | 8 | 128 | - |
| | #hard negatives | 7 | 1 | - |
| | Hard negative source | stage 1 teacher | stage 1 teacher | - |
| | #Epochs | 2 | 40 | - |
| **Self Training** | | | | |
| **Pre-train** | Learning rate | $1 \times 10^{-5}$ | $1 \times 10^{-5}$ | $1 \times 10^{-5}$ |
| | Batch size | 32 | 32 | 32 |
| | #hard negatives | 7 | 7 | 7 |
| | Hard negative source | stage 2 teacher | stage 2 teacher | stage 2 teacher |
| | #Epochs | 1 | 1 | 10 |
| **Finetune** | Learning rate | $5 \times 10^{-6}$ | $1 \times 10^{-5}$ | - |
| | Batch size | 8 | 128 | - |
| | #hard negatives | 7 | 1 | - |
| | Hard negative source | stage 2 teacher | stage 2 teacher | - |
| | #Epochs | 2 | 40 | - |

Table 8: Hyperparameter settings for retriever training.

| | Hyperparameters | MS-MARCO | Natural Questions | BEIR |
|---|---|---|---|---|
| **Shared** | Max length | 156 | 156 | 351 |
| | Optimizer | AdamW | AdamW | AdamW |
| | Scheduler | Linear warmup | Linear warmup | Linear Warmup |
| | Warmup proportion | 0.1 | 0.1 | 0.1 |
| | Weight decay | 0.1 | 0.1 | 0.1 |
| | #Params | 110M | 110M | 110M |
| **Teacher Reranker** | Learning rate | $1 \times 10^{-5}$ | $1 \times 10^{-5}$ | - |
| | Batch size | 12 | 12 | - |
| | #hard negatives | 40 | 15 | - |
| | Hard negative source | stage 2 teacher retriever | stage 2 teacher retriever | - |
| | #Epochs | 2 | 10 | - |
| **Self Training** | | | | |
| **Pre-train** | Learning rate | $1 \times 10^{-5}$ | $1 \times 10^{-5}$ | $1 \times 10^{-5}$ |
| | Batch size | 48 | 128 | 32 |
| | #hard negatives | 40 | 15 | 7 |
| | Hard negative source | stage 2 teacher retriever | stage 2 teacher retriever | stage 2 teacher retriever |
| | #Epochs | 1 | 1 | 5 |
| **Finetune** | Learning rate | $1 \times 10^{-5}$ | $1 \times 10^{-5}$ | - |
| | Batch size | 12 | 12 | - |
| | #hard negatives | 40 | 15 | - |
| | Hard negative source | stage 2 teacher retriever | stage 2 teacher retriever | - |
| | #Epochs | 2 | 10 | - |

Table 9: Hyperparameter settings for reranker training.

| Model (→) | BERT | | | Condenser | | | Contriever | | | coCondenser | | | RetroMAE | | |
|---|---|---|---|---|---|---|---|---|---|---|---|---|---|---|---|
| Dataset | GenQ | Tea-cher | Stu-dent | GenQ | Tea-cher | Stu-dent | GenQ | Tea-cher | Stu-dent | GenQ | Tea-cher | Stu-dent | GenQ | Tea-cher | Stu-dent |
| TREC-COVID | 66.3 | 60.4 | **67.9** | 64.2 | 65.3 | **67.6** | 58.0 | 46.2 | **62.1** | 68.4 | 74.1 | **76.7** | 70.0 | 70.0 | **74.1** |
| BioASQ | **33.3** | 25.7 | 30.1 | 29.7 | 24.6 | **32.0** | 34.9 | 30.6 | **35.2** | 31.0 | 29.5 | **35.7** | 40.6 | 39.0 | **43.0** |
| NFCorpus | **28.8** | 26.0 | 30.2 | **31.3** | 26.0 | 30.9 | 33.8 | 32.5 | **34.4** | 34.6 | 33.1 | **34.9** | 34.8 | 34.3 | **35.4** |
| NQ | 40.8 | 45.1 | **45.9** | 43.2 | 44.6 | **46.9** | 38.6 | 47.1 | **48.4** | 44.4 | 48.4 | **50.3** | 44.2 | 50.3 | **50.5** |
| HotpotQA | 52.5 | 53.0 | **55.1** | 53.3 | 52.5 | **56.2** | 58.7 | 62.2 | **62.8** | 56.5 | 56.9 | **58.4** | 59.0 | **62.3** | 62.1 |
| FiQA-2018 | 28.1 | 24.4 | **28.2** | 27.7 | 23.1 | **29.1** | 32.9 | 29.4 | **33.7** | 33.0 | 30.1 | **33.3** | 34.9 | 32.1 | 34.5 |
| Signal-1M (RT) | 26.2 | 24.8 | **27.0** | **28.3** | 27.3 | 27.3 | 27.3 | 26.7 | **28.4** | 27.0 | 27.8 | **29.5** | 27.8 | 29.0 | **30.5** |
| TREC-NEWS | 35.8 | 33.7 | **38.3** | 37.3 | 32.3 | **40.5** | 40.0 | 39.4 | **42.9** | 40.7 | 40.1 | **43.8** | 39.4 | 37.3 | **40.7** |
| Robust04 | 35.4 | 35.1 | **38.6** | 37.9 | 35.7 | **40.9** | 36.8 | 41.5 | **44.6** | 40.6 | 43.6 | **46.8** | 40.0 | 42.9 | **44.0** |
| ArguAna | **43.1** | 32.9 | 42.8 | 44.1 | 31.7 | **45.2** | 42.3 | 31.2 | **49.4** | 46.3 | 37.3 | **50.1** | 40.3 | 32.9 | **43.0** |
| Touché-2020 | 16.8 | **28.7** | 28.5 | 17.4 | 29.1 | **30.1** | 16.9 | 20.4 | 19.4 | 17.9 | 30.5 | **30.4** | 16.0 | **26.9** | 26.1 |
| CQADupStack | 31.9 | 26.0 | **32.3** | **32.7** | 27.7 | 32.3 | 33.2 | 32.4 | **36.1** | 35.5 | 32.6 | **35.8** | 35.5 | 33.8 | **37.2** |
| Quora | 83.8 | 82.5 | **84.6** | 85.6 | 84.9 | **85.8** | 84.5 | 84.0 | **86.1** | 86.1 | 85.6 | **86.8** | 85.3 | 85.2 | **86.2** |
| DBPedia | 32.8 | 32.5 | 32.0 | **34.7** | 34.0 | 33.9 | 37.4 | **40.4** | 38.7 | 36.3 | **37.5** | 37.2 | 35.4 | **37.7** | 35.7 |
| SCIDOCS | 14.1 | 12.4 | **14.6** | 14.3 | 12.6 | **15.5** | 15.8 | 14.8 | **17.0** | 15.2 | 14.3 | **16.1** | 16.1 | 14.7 | **16.9** |
| FEVER | 66.9 | 66.9 | **72.0** | 69.4 | 68.5 | **72.2** | 74.8 | 73.2 | **75.6** | 69.2 | 71.8 | **72.2** | 73.0 | 72.4 | **76.5** |
| Climate-FEVER | 22.2 | 18.4 | **22.3** | 20.2 | 14.7 | **22.3** | 23.4 | 17.9 | **23.6** | 21.8 | 17.6 | **22.2** | 19.7 | 17.2 | **23.8** |
| SciFact | **65.4** | 57.0 | 63.4 | **64.4** | 57.1 | 63.2 | 68.6 | 66.3 | **69.8** | **66.2** | 61.0 | 65.1 | 67.8 | 64.7 | **68.7** |
| Avg. Performance | 40.2 | 38.1 | **41.9** | 40.9 | 38.4 | **42.9** | 42.1 | 40.9 | **44.9** | 42.8 | 42.9 | **45.9** | 43.3 | 43.5 | **46.1** |

Table 10: Results on BEIR benchmark (nDCG@10) when adopting different pre-trained models. **Teacher** refers to the model finetuned on MS-MARCO.

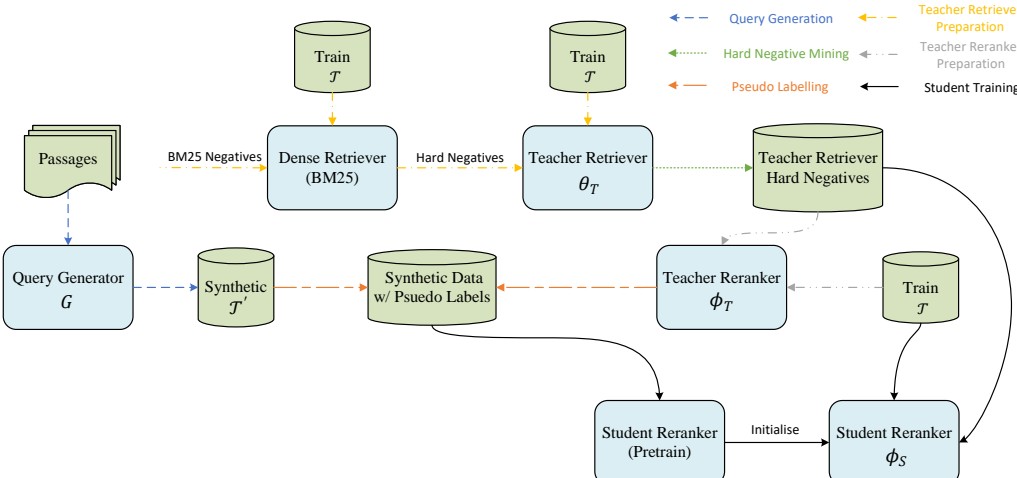

Figure 5: The overview of adapting the self-training framework to reranker.

| Motivation | | | |
|---|---|---|---|
| Practical ☐ | Cognitive | Intrinsic | Fairness |

| Generalisation type | | | | | |
|---|---|---|---|---|---|
| Compositional | Structural | Cross Task ☐ | Cross Language | Cross Domain ☐ | Robustness |

| Shift type | | | |
|---|---|---|---|
| Covariate | Label | Full | Assumed ☐ |

| Shift source | | | |
|---|---|---|---|
| Naturally occuring ☐ | Partitioned natural | Generated shift | Fully generated |

| Shift locus | | | |
|---|---|---|---|
| Train–test | Finetune train–test ☐ | Pretrain–train | Pretrain–test |