# OpenReview forum: "Noisy Self-Training with Synthetic Queries for Dense Retrieval"
_EMNLP/2023/Conference — EMNLP 2023 Findings_

### Official Review · Reviewer_ugd9 · 2023-08-03

**Soundness:** 3

**Excitement:**

3: Ambivalent: It has merits (e.g., it reports state-of-the-art results, the idea is nice), but there are key weaknesses (e.g., it describes incremental work), and it can significantly benefit from another round of revision. However, I won't object to accepting it if my co-reviewers champion it.

**Paper Topic And Main Contributions:**

The authors propose a self training strategy to iteratively improve the augmented training data. The experiments on MS MARCO, NQ and BEIR shows that the self training strategy improves over the standard training strategy for both bi-encoder and cross-encoder ranking models. Generally, I think this paper is a variant of GPL by replacing cross-encoder with bi-encoder teacher to improve the synthetic data generation throughput. As for the comparison of PTR, this work requires human label data while PTR does not.

**Questions For The Authors:**

1. Have you tried to train a single bi-encoder model with the synthetic data and cross-encoder scores provided by GPL? If self training is comparable or even better; then, we can further claim that the self training approach can produce equal outcome but saving a lot of time. Maybe give a rough data generation time estimation comparison is better.
2. Have you tried self training with sentence cropping as COCO-DR[1] or the variant of DRAGON[2]? In this way, the data generation process can be more efficient.

[1] Yue Yu, Chenyan Xiong, Si Sun, Chao Zhang, and Arnold Overwijk. 2022. Coco-dr: Combating distribution shifts in zero-shot dense retrieval with contrastive and distributionally robust learning. In Proc. EMNLP.
[2] Sheng-Chieh Lin, Akari Asai, Minghan Li, Barlas Oguz, Jimmy Lin, Yashar Mehdad, Wen-tau Yih, and Xilun Chen. 2023a. How to Train Your DRAGON: Diverse Augmentation Towards Generalizable Dense Retrieval. ArXiv:2302.07452

**Reasons To Accept:**

Overall, this paper is well motivated and experiments are comprehensive enough.
1. The motivation of the paper is clear and reasonable. Overall, the paper is easy to read.
2. The self training approach can improve both bi-encoder and cross-encoder ranking models and the authors demonstrate the effectiveness of the approach across various datasets.


**Reasons To Reject:**

The paper seems OK. However the noise induced by synthetic queries and possible solutions have been discussed by GPL and PTR. Thus, I feel the main contribution is to save training time compared to GPL; thus, I feel the contribution is not enough for a long paper.
1. I think the proposed method is very similar to Promptagator (PTR) by generating synthetic queries using seq2seq models and cleaning data with self consistency. And the proposed method is also similar to GPL but replacing the cross-encoder teacher with a bi-encoder teacher. The main difference is that PTR and GPL train one model for each dataset while the authors train one model for all datasets (but still have two models for MS MARCO and BEIR). Although the main claim that synthetic queries are noisy and the corresponding solution are reasonable, this claim and solution are well known according to the PTR and GPL. That is, I do not see any significant impact of the paper rather than just trying the different training settings of GPL.
2. As for comparison, I think that the authors should also follow GPL’s training strategies and train a single retrievers for all BEIR dataset. In this way, we can see the importance or the gain of self training. However, the authors only compare with the single model trained on GenQ.

**Reproducibility:**

4: Could mostly reproduce the results, but there may be some variation because of sample variance or minor variations in their interpretation of the protocol or method.

**Reviewer Confidence:**

4: Quite sure. I tried to check the important points carefully. It's unlikely, though conceivable, that I missed something that should affect my ratings.

---

> ### Author Rebuttal · Authors · 2023-08-29
>
> Thanks for your comments.
>
> ---
> > W1: Differences from GPL and PTR
>
> The main motivation of GPL and PTR is to improve the retrieval accuracy in out-of-domain in a zero- or few-shot manner. By contrast, our method can improve retrievers in both in-domain (rich resources) and out-of-domain (zero resources) settings. Moreover, GPL and PTR all rely on external models to gain improvements, either through soft label generation using an expensive reranker or a massive query generator. Instead, our method proves that a retriever can be self-improved. The merits of our method can not be ignored given the fact that it can achieve comparable results to GPL and PTR while being efficient in both training and model deployment.
>
> ---
> > Q1: Single GPL model
>
> | Model | FQ | SF | AA | CF | DB | CQ | QU | SD | FE | NF | TC | T2 | HP | NQ | RB | TN | SG | BA | Avg |
> | :--: | :--: | :--: | :--: | :--: | :--: | :--: | :--: | :--: | :--: | :--: | :--: | :--: | :--: | :--: | :--: | :--: | :--: | :--: | :--: |
> | GPL (66M * 18) | 34.4 | 67.4 | **55.7** | **23.5** | **38.4** | 35.7 | 83.6 | **16.9** | **75.9** | 34.5 | 70.0 | 25.5 | 58.2 | 48.3 | 43.7 | 42.1 | 27.6 | 44.2 | 45.9 |
> | GPL (110M) | **34.6** | **67.6** | 53.4 | 23.4 | 38.1 | **36.2** | 83.6 | 16.7 | 75.9 | 34.9 | 73.6 | 24.4 | 56.3 | 49.0 | **46.9** | 42.8 | 27.0 | **44.7** | **46.1** |
> | Student Retriever | 33.3 | 65.1 | 50.1 | 22.2 | 37.2 | 35.8 | **86.8** | 16.1 | 72.2 | **34.9** | **76.7** | **30.4** | **58.4** | **50.3** | 46.8 | **43.8** | **29.5** | 35.7 | 45.9 |
>
> We show the extra experiments when training a single GPL model. We follow the same training setting as described in GPL but only change the initialised checkpoint to our teacher retriever presented in the paper to ensure a fair comparison. As shown in the table,  the single GPL model achieves marginal improvements over ours. However, this comes at a substantial extra cost. The time required for soft labelling increases by roughly 20 times (~0.8 ms per query for ours while ~20 ms for reranker labelling) and the training time also triples. Thus, considering such factors, our method is still competitive.
>
> ---
> > Q2: Using crop sentences as query
>
> Yes, we tried using crop sentences in our preliminary experiments. In the in-domain settings (i.e., MSMARCO and NQ), we found the results of using crop sentences are inferior to those of using synthetic queries. We reason that since synthetic queries are sampled from a query generator trained on gold query-passage pairs, the distribution of synthetic queries and their formats are much closer to gold ones compared to sentence croppings. In the BEIR benchmark, we found sentence croppings lead to better performance in tasks where the query is in the format of short sentences (such as Climate-Fever for fact verification), while in other tasks where the query is natural questions, using synthetic queries is a better choice. To make the settings consistent and our method comparable to others, we used synthetic queries for all our experiments. Future work would consider how to mix both synthetic queries and sentence croppings to achieve better overall results.

---

### Official Review · Reviewer_uzg5 · 2023-08-03

**Soundness:** 4

**Excitement:**

3: Ambivalent: It has merits (e.g., it reports state-of-the-art results, the idea is nice), but there are key weaknesses (e.g., it describes incremental work), and it can significantly benefit from another round of revision. However, I won't object to accepting it if my co-reviewers champion it.

**Paper Topic And Main Contributions:**

This paper works on dense retrieval. The authors propose a self-training framework to improve dense retrieval with limited data. Specifically, they propose to generate queries for unlabelled passages. However, such generated dataset may carry noise, the authors propose also to generate pseudo labels by a teacher model to improve the quality of the generated dataset. The authors evaluate their method on both general domain and out-of-domain benchmarks. The results show the proposed method can not only improve the performance, but also reduce the need for labeled data.

**Questions For The Authors:**

1. What is the noise rate in $T’$? This information will be a strong evidence to motivate your method. Either a small scale human evaluation or automatic model evaluation is ok.

2. How did you evaluate the quality of the pseudo labels for generated queries? I suppose its quality should be higher than the original $T’$, otherwise the student model can hardly be improved.

3. What is the performance if we use hard labels in Algorithm 1 line 6?

4. Have you considered also updating the query generator via the self-training mechanism?


**Reasons To Accept:**

1. This paper is well written and easy to follow.

2. The idea of improving dense retrieval using self-training sounds reasonable to me.

3. The experiments are extensive and support the main claim.


**Reasons To Reject:**

1. The technical novelty of this paper is limited. But it is a minor issue for an empirical study paper.

2. The motivation of generating pseudo labels for synthetic queries is not strong enough. Even though the authors provide a motivation example in Table 1, a comprehensive noise analysis of the generated queries is favorable.


**Reproducibility:**

4: Could mostly reproduce the results, but there may be some variation because of sample variance or minor variations in their interpretation of the protocol or method.

**Reviewer Confidence:**

3: Pretty sure, but there's a chance I missed something. Although I have a good feel for this area in general, I did not carefully check the paper's details, e.g., the math, experimental design, or novelty.

---

> ### Author Rebuttal · Authors · 2023-08-29
>
> Thanks for your comments.
>
> ---
> > Q1: What is the noise rate in $\mathcal{T}'$?
>
> We did an automatic evaluation of the noise rate in $\mathcal{T}'$ in MSMARCO. We consider one query-passage pair as noisy if the originating passage cannot be ranked as the top-1 prediction by a well-trained retriever (the teacher retriever in our case). We find that only 22\% of query-passage pairs survive after this procedure, which means there exist other passages that are more semantically related than the originating passages for 78\% of synthetic queries. Moreover, considering the characteristics of MSMARCO where passages are split from documents, there still exist other passages that are semantically related to those surviving queries. In other words, simply using 0/1 hard labels by treating all other passages equally as negatives is not an ideal solution.
>
> ---
> > Q2: How did you evaluate the quality of the pseudo labels for generated queries?
>
> We first randomly sampled 100 synthetic queries and manually checked if the originating passage for each query was correct. We found 49 of them are correct. Out of these 49 queries, we found that 93% of the time the soft labels can find other passages that can also correctly answer the query, that is assigning a high probability mass to these passages. For the other 51 incorrect query-passage pairs, a different passage that makes a better match than the originating one can be found for 42 of them. This shows that soft labels are beneficial to both types of query-passage pairs.
>
> ---
> > Q3: What is the performance if we use hard labels in Algorithm 1 line 6?
>
> We show the experiment of using hard labels in the 3rd row of Table 4 (i.e.w/o pseudo labels). In both datasets, we see directly using the hard labels by taking the originating passage as the positive does not benefit the retriever, and it even hurts the results in most metrics. The corresponding experiment in the out-of-domain BEIR benchmark is the GenQ variant, we see that further training the teacher retriever on hard labels doesn’t provide any benefits in overall performance.
>
> ---
> > Q4: Have you considered also updating the query generator via the self-training mechanism?
>
> The effectiveness of our method mainly depends on two aspects: the quality of generated soft labels and how close the synthetic queries are to the gold ones. In this work, we mainly focus on the first one. However, it is interesting to see how to improve the quality of query generation. One way could be applying large language models but at the cost of massive inference time. Using the self-training framework is a good option by iteratively mixing gold queries with selected synthetic queries with high quality to train an improved generator. Another way could be taking the performance on the target task (i.e. retrieval) as a reward to improve the generator. We would like to explore this as future work.

---

### Official Review · Reviewer_E4NL · 2023-08-20

**Typos Grammar Style And Presentation Improvements:** N/A
**Soundness:** 3

**Excitement:**

3: Ambivalent: It has merits (e.g., it reports state-of-the-art results, the idea is nice), but there are key weaknesses (e.g., it describes incremental work), and it can significantly benefit from another round of revision. However, I won't object to accepting it if my co-reviewers champion it.

**Missing References:**

N/A

**Paper Topic And Main Contributions:**

This paper proposes a self-training framework that iteratively enhances the retriever. More specifically, the student retriever is improved continuously trained by the soft-label and negatives given by the teacher retriever. This final student retriever is evaluated on both in-domain and out-of-domain scenarios. Several ablation studies reveal how the model works.

**Questions For The Authors:**

Please check the "Reason to Reject" section, my questions are pretty much there.

**Reasons To Accept:**

- The proposed are evaluated from several perspectives in the experiment.

- The motivation to solve "out-of-domain" problems in retrieving is interesting.

- The paper is overall easy to follow.

**Reasons To Reject:**

- The main contribution of this paper is to use self-training framework to learn a student retriever that takes training data query-paraphrase pairs and the negatives given by a teacher retriever in an interactive manner. The authors claim that using the soft labels generated by teacher retrievers would be more robust to noise. However, this claim is not well supported or justified in the paper. More specially, since the teacher model is trained with the labelled data (in the beginning), why using the soft label can be more robust?

- Another main claimed strength of the proposed model is the generalizability to "out-of-domain". However, It is not clear how the model achieves this ability. There is not specific design in the training algorithm. Also, the out-of-domain results of student retriever in table 5 seem to be very weak compared to baselines, it is not convincing that the model can really solve "out-of-domain" scenarios.

- Since the proposed self-training is an iterative training framework, a convergence analysis (e.g., convergence experiments) is required to understand the training process.

**Reproducibility:**

3: Could reproduce the results with some difficulty. The settings of parameters are underspecified or subjectively determined; the training/evaluation data are not widely available.

**Reviewer Confidence:**

3: Pretty sure, but there's a chance I missed something. Although I have a good feel for this area in general, I did not carefully check the paper's details, e.g., the math, experimental design, or novelty.

---

> ### Author Rebuttal · Authors · 2023-08-29
>
> Thanks for your comments
>
> ---
> > W1: The claim that using the soft labels generated by teacher retrievers would be more robust to noise is not well-supported.
>
> We gave a motivating example in Table 1 to show why soft labels are more robust to potential noise compared to using 0/1 hard negatives. Since the query generator is not perfect, the synthetic queries may contain hallucinations or only reflect one aspect of the originating passages. Moreover, sometimes the synthetic query is only superficially related to the originating passages with only keywords overlapped. As shown in the example of Table 1, we find there exists more than one passage that is more semantically related to the synthetic query than its originating one, and this is not uncommon in the whole dataset. In such a case, simply taking the originating passage as positive with label 1 and treating others equally with label 0 will mislead the model during training. By contrast, soft labels provide the retriever with a larger number of training signals by learning to match a distribution. Thus, soft labels are generally more favorable to hard labels by providing the relevances to a query that is comparable among passages. We also empirically found that using hard labels can hurt the performance, as shown in the 3rd row of Table 4.
>
> ---
> > W2: Not clear how the model achieves “out-of-domain” ability
>
> We show that when given only the unlabelled text corpus from tasks in out-of-domain settings, a teacher model trained in in-domain data (e.g., wikipedia) can generate rich training signals for out-of-domain data (e.g., biomedical and scientific). Moreover, a student model trained on such silver training data can surpass the teacher in out-of-domain settings. That is to say, a model trained in in-domain data can produce a model with a significantly better performance in unseen domains without using any human-annotated supervision signals.
>
> ---
> > W3: The out-of-domain results of student retriever in Table 5 seems weak compared to other baselines
>
> The major baselines we compared are GPL and PTR. As shown in Table 5, PTR uses a massive language model for synthetic queries, while our method only employs a T5-base model with 220M parameters. Considering the comparable results, our method is more practical compared to theirs. Moreover, since the quality of their synthetic queries is expected to be much better than ours, it is reasonable to expect that by integrating with their queries, our method could gain further benefits. For GPL, we use similar settings for query generation. However, they used a more expensive reranker model to generate soft labels, which is 20 times longer than ours based on our rough comparison. Moreover, their training time is also roughly 3 times longer than ours. Therefore, considering the efficiency of our method in query generation, soft labelling, and training, our method is useful and its merits can not be overlooked by simply comparing the raw numbers presented in the table.
>
> ---
> > W4: Missing convergence analysis
>
> This was mentioned in lines 944-945 in Appendix A1 and sorry for not presenting this explicitly. We show the detailed convergence analysis below. As we can see, employing one iteration of our algorithm leads to the best results and the performance starts to decline with more iterations.
> | Iteration| MSMARCO |
> | :--: | :----: |
> | 0 | 37.4 |
> | 1 | **39.4** |
> | 2 | 39.1 |
>
>
> | Iteration | FQ | SF | AA | CF | DB | CQ | QU | SD | FE | NF | TC | T2 | HP | NQ | RB | TN | SG | BA | Avg |
> | :--: | :--: | :--: | :--: | :--: | :--: | :--: | :--: | :--: | :--: | :--: | :--: | :--: | :--: | :--: | :--: | :--: | :--: | :--: | :--: |
> | 0 | 30.1 | 61.0 | 37.3 | 17.6 | **37.5** | 32.6 | 85.6 | 14.3 | 71.8 | 33.1 | 74.1 | **30.5** | 56.9 | 48.4 | 43.6 | 40.1 | 27.8 | 29.5 | 42.9 |
> | 1 | 33.3 | 65.1 | 50.1 | **22.2** | 37.2 | 35.8 | **86.8** | 16.1 | **72.2** | **34.9** | **76.7** | 30.4 | **58.4** | **50.3** | **46.8** | **43.8** | **29.5** | 35.7 | **45.9** |
> | 2 | **33.7** | **66.7** | **50.6** | 22.2 | 35.8 | **36.5** | 86.2 | **16.5** | 70.9 | 34.9 | 75.7 | 29.2 | 57.5 | 49.2 | 46.4 | 42.8 | 28.7 | **36.9** | 45.6 |

---

### Meta-Review · Area_Chair_SdsZ · 2023-09-27

**Recommendation:** 3

**Metareview:**

This paper proposes a self-training strategy to iteratively improve the augmented training data. Compared with baseline models, the proposed method can not only improve the performance, but also reduce the need for labeled data. The reviewers generally hold a positive view of this paper. The author has also addressed some of the concerns raised by the reviewers. However, further careful revision and supplementation may still be required. The rebuttal has addressed reviewers' concern on the motivation and novelty concerns, and the authors agree to revise this paper accordingly.

---

### Decision · Program_Chairs · 2023-10-07

**Decision:**

Accept-Findings

**Comment:**

This paper proposes a self-training strategy to iteratively improve the augmented training data. Compared with baseline models, the proposed method can not only improve the performance, but also reduce the need for labeled data. The reviewers generally hold a positive view of this paper. The author has also addressed some of the concerns raised by the reviewers. However, further careful revision and supplementation may still be required. The rebuttal has addressed reviewers' concern on the motivation and novelty concerns, and the authors agree to revise this paper accordingly.